# Comparative Studies of *Fraxinus* Species from Korea Using Microscopic Characterization, Phytochemical Analysis, and Anti-Lipase Enzyme Activity

**DOI:** 10.3390/plants9040534

**Published:** 2020-04-20

**Authors:** Kazi-Marjahan Akter, Woo Sung Park, Hye-Jin Kim, Atif Ali Khan Khalil, Mi-Jeong Ahn

**Affiliations:** 1College of Pharmacy and Research Institute of Pharmaceutical Sciences, Gyeongsang National University, Jinju 52828, Gyeongsangnam-do, Korea; marjahan7silva@gnu.ac.kr (K.-M.A.); pws14093@gnu.ac.kr (W.S.P.); black200203@gnu.ac.kr (H.-J.K.); Atif.ali@numspak.edu.pk (A.A.K.K.); 2Department of Biological Sciences, National University of Medical Sciences, 46000 Rawalpindi, Pakistan

**Keywords:** *Fraxinus* species, microscopic characterization, chemical profiles, anti-lipase activity

## Abstract

*Fraxinus* species belongs to the Oleaceae family, commonly known as Ash tree, and has been utilized as a folk medicine with various medicinal properties, including anti-obesity activity. The goal of the present study was to establish quality control parameters using microscopic characterization, phytochemical differentiation, and anti-lipase activity evaluation of five *Fraxinus* plants in Korea. Microscopic evaluation of the lower surface, petiole, and midrib of leaves, and stem bark showed discriminative anatomical characteristics, such as the stomatal index of the lower leaf surface; the number of sclerenchyma cells, and the diameter of parenchyma cells in the petiole and midrib; and the cork cell size and fiber frequency in the stem bark. Phytochemical analysis using high-performance liquid chromatography revealed the significant variation in the chemical profiles of the 12 major secondary metabolites among the samples. The orthogonal projections to latent structure-discrimination analysis efficiently differentiated each group belonging to each *Fraxinus* plant with the anatomical and quantification data. *F. rhynchophylla* and ligstroside showed the most potent anti-lipase activity among the plants and the 12 major metabolites, respectively. These findings could serve as the scientific criteria for the appropriate identification and establishment of standards for the use of *Fraxinus* species as medicinal plants.

## 1. Introduction

The genus *Fraxinus* belongs to the Oleaceae family, members of which are commonly known as Ash trees, which consist of 43 species, and are widely distributed throughout North America, Europe, and eastern and western Asia [1,2,3,4]. This genus is monophyletic and distinctive in the Oleaceae family by mostly having comparatively large imparipinnate leaves and one-seeded samaras [5]. Four species and one variety of the *Fraxinus* genus can be found in Korea: *Fraxinus chiisanensis* Nakai (FC), *F. mandshurica* Rupr. (FM), *F. rhynchophylla* Hance (FR), *F. sieboldiana* Blume (FS), and *F. sieboldiana* var. *angustata* Blume *(*FSV) [6]. *Fraxinus* species have been used as folk medicine in various places worldwide for their diuretic and mild laxative effects as well as for treating inflammation, constipation, arthritis pain, itching, and cystitis, etc. [7]. A traditional oriental drug called “Qin pi” (Cortex Fraxini) is defined as dried bark of the stem or branches of the Korean ash tree, *F. rhynchophylla* (Mul-pu-re-na-mu), which is used as a diuretic, analgesic, astringent, antirheumatic, and antiphlogistic agent [8,9]. The dried bark of FM is sold as an alternative for Cortex Fraxini and is widely used in China, South Korea, and Japan [10,11]. The leaf extract of FM is an immunosuppressant that has been found to act via the inhibition of IL-2 and IgE production in mouse spleen cells and U266 cells [10].

A series of chemical constituents, including coumarins, secoiridoids, phenylethanoids, lignans, and flavonoids have been isolated from *Fraxinus* species [12]. Notably, several bioactive compounds have been identified in *Fraxinus* species that possess anticancer, anti-inflammatory, immunomodulatory, antioxidant, anti-obesity, anti-microbial, neuroprotective, hepatoprotective, anti-hypertensive, anti-aging, and anti-allergic activities [13,14].

Using only a single method, identification of plants can be challenging if the samples are from an area of sympatry in which interspecific hybridization may occur, or because of misidentification or mislabeling in arboreta or herbaria, or synonymy of taxa. Therefore, it is necessary to determine the exact botanical origin to get the correct biological effect. There are several authentication methods in order to identify the plant products for quality and safety confirmation, and all analytical methods are complementary to each other. Microscopic observation is exceptionally valuable for the evaluation of dried plant materials whose characteristics can be altered from that of the fresh state. Moreover, it has the benefit of low instrumental and effective costs, and has been strictly applied in the established Pharmacopoeia [15,16,17,18]. Chemical analysis of crushed samples or extracts through high-performance liquid chromatography (HPLC) or thin-layer chromatography are reliable and excellent methods for the authentication of medicinal plant species and their products, particularly when morphological authentication is not possible. Nevertheless, it can be a difficult task to unequivocally identify all peaks in the chromatograms using HPLC with retention times, spectra, and multiple internal standards [19]. Molecular techniques are convenient, highly specific, relatively quick, and straightforward processes, but they are expensive and require special training or experience [20]. Although outer morphological and phylogenetical analyses of *Fraxinus* spp. have been reported, there is no organized analysis of the comparative inner morphological study and the quantification of chemical constituents, for which the identification of the botanical origin of *Fraxinus* spp. has been assessed.

Meanwhile, FR and its isolated compounds were reported to possess anti-obesity characteristics via inhibition of pancreatic lipase [21,22]. Pancreatic lipase is the major enzyme that converts ingested triglycerides to fatty acids, and as such decreases fat absorption through pancreatic lipase inhibition and can be useful to treat obesity [23,24]. Many countries have used medicinal plants as dietary supplements for the controlling of body weight and, thus, natural inhibitors of pancreatic lipase could be potential candidates to control obesity [25].

Therefore, this study was designed to use systematic microscopic observations and secondary metabolite quantification through HPLC to establish the standard parameters for anatomical and chemical differentiation of the four *Fraxinus* species and one variety that have been utilized as a folk medicine in Korea. In addition, the goal of our study was to evaluate the anti-lipase activity of the five *Fraxinus* plants.

## 2. Results

### 2.1. Anatomical Characteristics of the Leaf

Regarding the surface view of the leaf blade, the upper (adaxial) surface of leaves from all samples showed highly similar anatomical features among the five plants. Taxonomically important anatomical characteristics were only found on the lower surfaces of leaves. The epidermis consisted of irregularly shaped cells with a sinuate pattern on the lower (abaxial) surface, which presented a few number of unicellular trichomes. Nine to 11 epidermal cells were organized in a rosette around the trichome bases (Figure 1a). Secretory glandular trichomes were observed on the lower surface of all the samples. FSV had the highest number of glandular trichomes with the mean value of 7.0 ± 0.4 in a 500 × 500-μm^2^ area and the smallest diameter of glandular trichomes among the samples with the value of 32.3 ± 1.5 μm (Table 1). Different number of epidermal cells were found among the samples. FM and FSV displayed a greater number of epidermal cells with a mean value of 148.4 ± 12.5 and 122.8 ± 8.2 in a 300 × 300-μm^2^ area, respectively, followed by those of FR, FS, and FC with values ranged from 78.4 ± 11.3 to 98.5 ± 14.2 in the same area. The leaves of all samples were hypostomatic, and anomocytic stomata were found only on the lower surface with varying number of stomata, stomatal length and width, and stomatal index. FM exhibited the highest number of stomata with a mean value of 33.9 ± 4.0 in a 300 × 300 μm^2^ followed by FSV, FR, FS, and FC with the values ranging from 11.7 ± 0.4 to 24.6 ± 3.0 in the same area. The largest stomata was exhibited by FC (37.5 ± 2.0 μm length and 26 ± 2.4 μm width), whereas FM had the smallest stomata (22.8 ± 0.6 μm length and 17.1 ± 0.5 μm width). FM, FR, and FSV had a higher stomatal index than did the other two species, FC and FS (Table 1) (Figure 1b).

### 2.2. Anatomical Characteristics of the Petiole

As shown in Figure 2a, the petiole comprised several anatomical elements: a cuticle, epidermis, ground tissues of parenchyma, collenchyma, and sclerenchyma cells, conductive tissues of xylem and phloem, starch grains, and parenchymatous pith etc. The petiole of almost all samples was oval-shaped with a flat adaxial side and altered by two lateral wings, defining a wide and deep ditch. On the two lateral wings of the adaxial portion, there were conductive tissues with a very few sclerenchyma cells at the phloem periphery (Figure 2b). The epidermis had a single line of isodiametric cells with thin cuticles, covered with various numbers of multicellular and rarely unicellular trichomes. FC and FR exhibited larger epidermal cells at the adaxial portion than did other species. Two to three layers of annular-type collenchyma cells in the cortex were observed beneath the epidermis in all samples. FC had a lower ratio of cortex than did the other plants, with the mean value of 0.23 ± 0.03 μm (Table 2). A larger diameter of parenchyma cells in both (adaxial and abaxial) portions occurred in FC with the mean values of 35.3 ± 2.5 and 51.6 ± 4.6 μm, respectively. Sclerenchyma cells were observed at the periphery of the phloem region for all samples with substantial variation. FSV exhibited the highest number of sclerenchyma cells with a mean value of 80.3 ± 3.4 in an area of 300 × 300 μm^2^, followed by FR with 70 ± 4.7 in the same area. FM and FS displayed similar numbers of sclerenchyma cells with the mean values of 51 ± 3.2 and 54.9 ± 3.0, respectively, and FC had the lowest number of sclerenchyma cells. The stele (phloem and xylem) was made up of numerous libero-ligneous fascicles of collateral type and provided a ring of vascular bundles. Three different types of xylem vessels, consisting of annular, spiral, and pitted vessels, were observed in all samples (Figure 2c). A thin-walled parenchymatous pith region was located at the center of the stele. Moreover, considerable similarity and dissimilarity was occurred in the diameter of parenchyma cells in the pith region. The parenchyma cells of FC had the largest diameter of 81.3 ± 2.6 μm, whereas those of FS and FSV had the smallest diameter in the range of 28.6–40.4 μm (Figure 2b) (Table 2).

### 2.3. Anatomical Characteristics of the Midrib

In the transverse section, the midrib was prominent on the abaxial side and the flat ditch was noticeable on the adaxial side for all samples (Figure 3). The mesophyll was dorsiventral and consisted of single or double-layered elongated palisade parenchyma on the upper side and multilayer of lacunose (spongy) tissue on the lower side. The single-layered palisade parenchyma was exhibited by the FC, FS, and FSV, whereas FM and FR had a double layer of that parenchyma. Unicellular, multicellular, and rarely secretory trichomes were observed on the surface of the midrib in all samples. The epidermis of the midrib was formed by a solo line of rectangular to squarish cells. FS and FSV had smaller epidermal cell than did the others (Table 3). The number of sclerenchyma cells displayed a substantial variation in the periphery of the phloem region for all samples. The highest number of sclerenchyma cells was observed in FSV with a mean value of 114.8 ± 18.7 in 300 × 300 μm^2^, followed by FS and FR with 75.5 ± 17.1 and 70.1 ± 6.5 in 300 × 300 μm^2^, respectively. FC had the lowest number with a mean value of 38.6 ± 3.0 in 300 × 300 μm^2^. Large variation among the samples was found for the diameter of parenchyma cells in the abaxial portion and pith region also. FC exhibited the largest diameter of parenchyma cells in the abaxial portion and in the pith region with the mean values of 53.4 ± 2.5 μm and 75.7 ± 3.7 μm, respectively. The small diameters of parenchyma cells were found in the abaxial portion of FSV and FS, as well as in the pith region also. No significant differences were found among the samples for the size of palisade cells and the palisade ratio (Figure 3) (Table 3).

### 2.4. Anatomical Characteristics of the Stem Bark

In the cross-section of stem bark, suberized cork layers occurred under the epidermis. Cork cells were differed in size among the studied plants. The larger cork cells were observed in the FC, FM, and FSV and the smallest cells were in the FR (Table 4). A number of parenchyma cells were observed in the cortex zone as well as the periderm zone. Phloem fibers appeared in a single or grouped form (Figure 4a). Considerable differences were observed in the number of phloem fibers. The highest number of phloem fibers was exhibited by the FS with a mean value of 138.4 ± 3.5 in 500 × 500 μm^2^, followed by FR with 123.0 ± 16.5 and FSV with 122.7 ± 4.2, respectively. FC and FM had the lowest number of phloem fibers (Table 4). A clustered continuous line of sclereids was formed in the stem bark of FM, FS, and FSV after the development of the parenchymatous cortex (Figure 4a). Brachysclereids (stone cells) were exhibited by all samples, except FC (Figure 4b). They occurred singly or in groups or patches in the cortex zone. Idioblastic sclereids appeared only in FM (Figure 4c). It was very difficult to differentiate between axial parenchyma and ray parenchyma cells in phloem zone because the arrangement of cells was extremely disordered. Secondary phloem rays were uniseriate to multiseriate. Uniseriate rays consisted of upright cells, whereas multiseriate rays had procumbent and square cells, as well as upright cells forming one to three marginal rows (Figure 4a). FSV and FM had a higher number of phloem rays, 4.8 ± 0.2 and 4.5 ± 0.5, respectively, than did the others. Although there were no significant differences in the size of phloem rays, FC displayed the largest phloem rays among the samples (Table 4).

### 2.5. HPLC-DAD Profiles of Five Fraxinus Plants

The HPLC method was employed to quantify and compare the major chemical constituents among the 100% methanol extracts of the stem bark of the four *Fraxinus* spp. and one variety collected during summer and autumn seasons. Twelve major peaks (**1**–**12**) were detected gradually at the retention times of 10.1, 13.4, 15.4, 15.9, 17.9, 21.1, 27.6, 28.6, 29.6, 30.8, 34.7, and 39.0 min (min) on HPLC chromatograms (Figure 5a,b).

The chemical constituents consistent to major peaks were determined to be six coumarins of esculin (**1**), isofraxidin-7-*O*-*β*-D-glucopyranoside (**3**), fraxin (**4**), mandshurin (**5**), fraxetin (**6**) and fraxinol (**10**); three phenylethanoids of tyrosol (**2**), calceolarioside A (**7**), and calceolarioside B (**8**); two secoiridoids of oleuropein (**11**) and ligstroside (**12**); and a lignan, pinoresinol-4′-*O*-*β*-D- glucopyranoside (**9**) (Figure 6) (Appendix A). These compounds have been isolated from the stem bark of *F. mandshurica* by our previously reported methods and used for the quantification as external standards [26].

Different quantities of these constituents were determined among all sample extracts from five *Fraxinus* plants collected during summer and autumn (Figure 5 and Table 5). Compounds **7** and **12** were detected in all extracts except that of FR from the autumn season, whereas compounds **1**, **8**, and **11** were detected in all extracts except FC from both seasons. In particular, FR, FS, and FSV exhibited more than 10 times as much as contents of compound **1** than did FM. Higher contents of compound **4** in the range of 21.8 to 65.6 mg/g·dry weight (DW) occurred in FS and FSV for both seasons than those of the others, ranging from 0 to 17.0 mg/g·DW. Among the 12 secondary metabolites, only two compounds **7** and **12** were determined in FC. The content of compound **12** was the highest in FC for both seasons, followed by that of FM. Compounds **2**, **5**, and **10** were detected only in FM with the mean values from 0.03 to 0.67 mg/g·DW. The content of compound **9** was prominently high in FM. Compound **3** was only found in FM with 2.39 ± 0.41 and 1.66 ± 0.14 mg/g·DW and in FR with 0.23 ± 0.04 and 0.22 ± 0.01 mg/g·DW, respectively, for the summer and autumn seasons. In general, the summer season stem bark showed the higher contents of individual compounds individually than did those of the autumn season.

### 2.6. Orthogonal Projections to Latent Structures-Discriminant Analysis (OPLS-DA) Multivariate Statistical Analysis

A multivariate analytical approach was chosen to evaluate and characterize anatomical and phytochemical differences and similarities among the four *Fraxinus* species and one variety. OPLS-DA multivariate statistical analysis was applied to the anatomical data of leaves and stem bark, and to HPLC quantification data of the 12 major peaks (Figure 7). The anatomical data for leaves in the OPLS-DA scores plot showed that the five separated groups were clustered but FR and FS were nearly overlapped. Scores for FM and FC were clustered at a distance from other plant groups for factors, such as the number, length, and width of the stomata of leaf lower surface, and the number of sclerenchyma cells of midrib and petiole (Figure 7a). The OPLS-DA scores plot with the anatomical data of stem bark decreased the distance of the FM group from the scores of other plant groups (Figure 7b). Application of the OPLS-DA on the combined anatomical data from leaves and the stem bark showed a clearer separation of each species and variety. No overlap was observed among the species and variety (Figure 7c). Moreover, the multivariate statistical analysis efficiently discriminated and classified into each group of *Fraxinus* species or variety when it combined HPLC quantification data with all anatomical data from leaves and stem bark (Figure 7d). FM and FC were significantly separated from the other three plants. FR, FS, and FSV appeared closer than the separated group in all cases. FR was near to FS, and FS was near to FSV.

### 2.7. Anti-lipase Activity

The pancreatic lipase inhibitory activity of the five *Fraxinus* plant extracts, fractions, and 12 compounds from the FM bark was determined. Porcine pancreatic lipase enzyme was used with orlistat as a positive control and *p*-NPB as the substrate. 

FR exhibited the most potent anti-lipase activity among the five *Fraxinus* plants with a percent inhibition of 46.7 ± 2.6%, followed by FM with 37.8 ± 1.3% inhibition at a concentration of 100 µg/mL (Figure 8a). Among the fractions of FM, the EtOAc fr. exhibited the highest lipase inhibitory activity with 44.2 ± 2.3% (Figure 8b).

Among the isolated compounds, ligstroside (**12**) had the most potent anti-lipase activity of 61.1 ± 1.7%, followed by calceolarioside B (**8**) (56.6 ± 0.9%) and calceolarioside A (**7**) (55.6 ± 2.3%) at the concentration of 100 μM (Figure 9a). Mandshurin (**5**) and esculin (**1**) showed 40% to 50% of inhibitory activity, while isofraxidin-7-*O*-*β*-D-glc (**3**), pinoresinol-4′-*O*-*β*-D-glc (**9**), tyrosol (**2**), fraxetin (**6**), and fraxinol (**10**) exhibited 30 to 40% inhibition (Figure 9a). Fraxin (**4**) and oleuropein (**11**) demonstrated weak inhibitory activity against the porcine lipase enzyme, which was less than 30% at the same concentration. Calceolarioside A (**7**), calceolarioside B (**8**), and ligstroside (**12**) also showed a dose-dependent pancreatic lipase inhibition at 25, 50, and 100 μM (Figure 9b).

## 3. Discussion

Authentication of plant resources, especially for medicinal plants, is an important issue when the plant material is to be directly used for medical treatment. Several authentication methods have been tried to identify the plant products for quality and safety [27]. They can help avoid mishaps because of the misidentification of materials anywhere during the production process from the collection of the raw material to the finished product. Therefore, in this study a systematically comparative method was built for four *Fraxinus* species and one variety from Korea by microscopic observation and HPLC profiling.

The results demonstrated similar and dissimilar anatomical characteristics of the leaves and stem barks by the comparative microscopic observations. All studied samples were dorsiventral and hypostomatic with an anomocytic type of stomata, but the number of stomata, stomatal size, and stomatal index were different for each *Fraxinus* species and the variety. The lower surface of leaves exhibited some useful parameters for distinguishing the studied samples. Although FC had the lowest frequency of glandular scales, epidermis cells, and stomata on the lower surface of leaves, FM exhibited the highest frequency of epidermis cells and stomata. FSV displayed the highest frequency of glandular trichomes on the lower leaf surface and the highest number of sclerenchyma cells in the petiole and midrib among the samples. Although single-layered palisade parenchyma cells of the midrib were found in FC, FS, and FSV, double-layered palisade parenchyma cells were found in the FM and FR. The number of phloem fiber in the stem bark of FR, FS, and FSV was much higher than in the stem bark of FC and FM.

Secondary metabolite profiling revealed considerable differences in the content of 12 major chemical constituents among the samples. While only two metabolites of calceolarioside A (**7**) and ligstroside (**12**) were determined in FC, tyrosol (**2**), mandshurin (**5**), and fraxinol (**10**) were detected in FM only. The contents of compounds **5** and compound **9** were prominent in FM. FC is an endemic tree species, only found in southwestern Korea and this species was categorized by the IUCN (International Union Conservation of Nature) red list as an endangered tree species [28,29,30]. In general, because the outer morphology of FC and FM trees is quite similar, it is often misidentified in the local area. However, they exhibited a much larger difference in their phytochemical profiles each other with characteristic inner morphological aspects in this study. The content of a coumarin compound, esculin (**1**) was much higher in FR, FS, and FSV than that of FM. A higher amount of another coumarin, fraxin (**4)** was shown by the FSV and FS in both seasons, which could be used as a valuable marker to distinguish them from the other three species. The total coumarin content was higher in FR, FS, and FSV, while the total secoiridoid content was higher in FC and FM. The existence of hydroxycoumarins and oleoside-type secoiridoids has been considered a typical feature of *Fraxinus* species [12]. The six coumarins were not detected in FC from both seasons but the content of a secoiridoid ligstroside in FC was the highest among the five *Fraxinus* plants. FS and FSV showed the highest total content of the 12 secondary metabolites, and followed by FM, FR, and FC in order. The content of secondary metabolites with biological activity is important for the use of medicinal plants, and the content varies according to seasons. The contents in plant bark is generally known to be higher in the bark collected in summer than in autumn. The distribution of FS and FSV scores located near each other in the OPLS-DA plot and created their groups, whereas scores of FS clustered between the FR and FSV. The scores for FM were clustered at a significant distance from other plants in all plots, and FC was consistently in the specific zone between the two plots in a separated group of OPLS-DA multivariate statistical analysis. Finally, our OPLS-DA multivariate statistical analysis successfully differentiated and classified each group of *Fraxinus* species and variety with the combination of HPLC quantification data and all the anatomical data for leaves and stem bark (Figure 6).

Because the occurrence of obesity remains high, the demand for effective and safe anti-obesity drugs as well as natural products are increasing. Despite several medications that have been used for the management of obesity over the years, most were withdrawn from the market due to serious adverse effects [31,32,33]. The presence of secondary metabolites, such as secoiridoids, polyphenols, flavonoids, lignans, and saponins contributes to the inhibition of the pancreatic lipase enzyme [21,22,23,25,34]. Several secoiridoids have been isolated from FR, which include bioactive hydroxyframoside B, ligstroside, oleuropein, and 2″-hydroxyoleuropein that possess anti-lipase activity *in vitro* [21]. A secoiridoid, oleuropein and a phenylethanoid, tyrosol also exhibited the anti-obesity effects for HFD-induced-fed mice [35,36]. Therefore, the anti-lipase activity of the stem bark of the five *Fraxinus* plants was employed to evaluate the anti-obesity effect in vitro. The results revealed that the studied *Fraxinus* plants exhibited the mild to moderate inhibitory effects against porcine pancreatic lipase. The EtOAc fraction of the stem bark of FM exhibited the highest anti-lipase activity among the fractions. All 12 isolates from FM bark showed the inhibitory potential against the lipase enzyme to some extent. Although the anti-lipase activity of compounds **2**, **9**, **11,** and **12** was previously reported, this is the first report on that of the other isolated compounds [21,22,37]. A secoiridoid, ligstroside (**12**) showed the most potent anti-lipase capacity among the 12 secondary metabolites, followed by two phenylethanoids of calceolarioside A (**7**) and B (**8**). Meanwhile, another secoiridoid, oleuropein (**11**) with a catechol moiety, showed a very weak inhibitory activity. Two phenylethanoid glucosides of compounds **7** and **8**, both with one caffeoyl moiety on an additional glucose group, showed more potent inhibitory activity than a simple phenylethanoid, tyrosol (**3**). Compounds **7** and **8** showed no significant difference in the inhibitory activity each other. The results suggest that a caffeoyl or a glucose moiety could play a role in the inhibitory activity of the compounds, and the position of the caffeoyl moiety on the glucose would not affect the activity. Phenylethanoid glycosides from *Cistanche phelypaea* displayed a certain inhibition activity on monoacylglycerol lipase [38]. Among the six coumarins, mandshurin (**5**) with a glucose moiety at the position 6 showed the most potent inhibitory activity, whereas fraxin (**4**) with the moiety at the position 8 displayed weaker inhibitory activity than fraxetin (**6**) without glycosylation. A lignan, pinoresinol-4′-*O*-*β*-D-glucopyranoside (**9**), also showed mild inhibitory activity.

## 4. Materials and Methods

### 4.1. Chemicals and Reagents

Glycerin (Junsei Chemical Co., Ltd.), Eau de Javel solution (Sigma, MN, USA), methylene blue (Samchun Pure Chemical Co., Ltd.), and ethanol (Daejung Chemicals & Metals Co., Ltd., Korea.) were used for the making of anatomical sample specimen. HPLC grade solvents (Fisher Scientific, Korea Ltd., Korea) were used in the HPLC analysis. Porcine pancreatic lipase (Type II), orlistat, Tris-HCl, Tris-base, morpholinepropanesulfonic acid (MOPS), *p*-nitrophenyl butyrate (*p*-NPB), and *N*,*N*-Dimethylformamide were purchased from Sigma-Aldrich Co. (St Louis, MO, USA). Ethylenediaminetetraacetic acid (EDTA) was purchased from Yakuri Pure Chemical Co., Ltd. (Kyoto, Japan) for measurement of pancreatic lipase activity assay. All reagents were biochemical reagent grade.

### 4.2. Plant Materials

Four *Fraxinus* species and one variety were collected from different geographic locations (Table 6) in Korea from June 2016 to November 2019 during the summer and autumn seasons. The samples were identified by Professor Mi-Jeong Ahn, College of Pharmacy, Gyeongsang National University, and the voucher specimens (PGSC No. 560.1 – PGSC No. 564.15) (Appendix A) were deposited in the herbarium of the College of Pharmacy, Gyeongsang National University.

### 4.3. Anatomical Analysis

Approximately the same aged, healthy, and well-acclimatized samples (lower surface, petiole, and midrib of the leaves, and stem bark) were obtained from the four *Fraxinus* spp. and one variety. More than eight pieces of the leaf lower surface, middle part of the petiole and, the lower part of the midrib were taken for the measurement. Transverse or vertical sections were made using a hand slicer. Eau de Javel solution was used for bleaching samples. Properly prepared material was mounted in glycerinated water (50%) on slide glass. The images were taken using a photomicroscope, Olympus BX53 (Olympus, Japan) connected to a camera (PixeLINK, ON, Canada) with image processing software (IMT i-Solution Inc., BC, Canada). More than 10 regions were measured for each photomicrograph, and the mean value was chosen as the representative for a specimen. More than three specimens were used to provide representative data for each plant.

### 4.4. Sample Preparation 

Dried bark or leaves of collecting plants were crushed in a mixer separately. Two grams of each sample was extracted thrice with 75 mL of 100% methanol under sonication for 90 min, each. The extract was filtered with a filter paper (Whatman No. 2, USA). The final volume was adjusted to 75 mL with 100% methanol and filtered through a 0.45 μm PTFE syringe filter (Whatman, New York, NY, USA) before HPLC analysis. The filtrate was dried under a stream of N_2_ gas and then used in pancreatic lipase assay. The air-dried bark of FM (960 g) was ground and extracted in sonication with 100% methanol. The extract was filtered and concentrated in vacuo. The resulting crude extract (187 g) was suspended in water and fractionated with *n*-hexane, dichloromethane, ethyl acetate and *n*-butanol, successively to yield *n*-hexane fr. (6.4%, w/w), CH_2_Cl_2_ fr. (3.2%), EtOAc fr. (32.1%), *n*-BuOH fr. (47.1%), and aqueous fr. (7.5%) fractions, respectively [26].

### 4.5. HPLC-DAD Analysis

An Agilent 1260 series HPLC system prepared with an autosampler, a column oven, a binary pump, and a degasser (Agilent Technologies, Palo Alto, CA, USA) was used. An aliquot (10 μL) of the standard or sample solution was directly injected on a Phenomenex Luna column (4.6 × 250 mm, 5 μm) with a companionable guard column. The gradient elution using an acetonitrile and water mixture was as follows: 10% acetonitrile to 15% for the first 15 min; 15% acetonitrile to 30% for the next 20 min; 30% acetonitrile to 40% for another 5 min; 40% acetonitrile to 100% for further 10 min. A conditioning phase was then used to return the column to the initial state for 5 min. The flow rate was 1 mL/min and column temperature was 30°C. The eluent was detected at 210 nm with a diode array detector (DAD). The chemstation software (Agilent Technologies) was used to operate this HPLC–DAD system.

### 4.6. Pancreatic Lipase Inhibition Assay

Inhibitory activity against pancreatic lipase was measured by a slightly modified method of previous report [39]. Briefly, an enzyme buffer was prepared by the addition of a solution of porcine pancreatic lipase, 2.5 mg/mL in buffer containing 10 mM MOPS (morpholinepropanesulfonic acid) and 1 mM EDTA (pH 6.8) to 169 μL Tris buffer (100 mM Tris-HCl and 5 mM CaCl_2_, pH 7.0). Then, either 20 μL of extract solution at the test concentration (100 μg/mL) or compound solution (100, 50 and 25 μM) or orlistat (1.0, 0.2, 0.04, 0.008, and 0.0016 μM) was mixed with 178 μL enzyme buffer and incubated for 15 min at 37°C with 2 μL of the substrate solution [10 mM *p*-NPB (*p*-nitrophenyl butyrate) in dimethylformamide]. The enzymatic reactions were allowed to proceed for 20 min at 37°C. Lipase activity was determined by measuring the hydrolysis of *p*-NPB into *p*-nitrophenol. An increase in light absorption at 405 nm was measured using a microplate reader (Synergy H1; BioTek Instruments, Inc., Winooski, VT, USA). Inhibition of lipase activity was expressed as the percentage decrease in optical density when porcine pancreatic lipase was incubated with the test compounds. Orlistat was used as a positive control. Lipase inhibition (%) was calculated according the following formula:Inhibition (%) = 100 − [(B-b)/(A-a)] × 100(1)
where A is the absorbance of incubated solution with substrate, a is the absorbance incubated solution with substrate and lipase, B is the absorbance of incubated solution with sample, substrate and lipase, b is the absorbance of incubated solution containing sample and substrate.

### 4.7. Statistical Analysis

All data were expressed as mean ± SD (standard deviation). One way ANOVA (Analysis of variance) followed by Tukey’s multiple comparison test was used for the statistical analysis using SPSS (Statistical Package for the Social Sciences) software (Version 16). Values of *p* < 0.05 were considered statistically significant. Orthogonal Projections to Latent Structures-Discriminant Analysis (OPLS-DA) was accomplished for multivariate analysis with software SIMCA (Ver.13, Umetrics, Sweden).

## 5. Conclusions

In conclusion, there were remarkable differences in the microscopic observations and phytochemical profiles of the four *Fraxinus* species and one variety. The microscopic data showed discriminative characteristics with regard to parameters of the lower leaf surface, petiole, and midrib of leaves and stem bark. HPLC chemical profiling exhibited considerable differences in the chemical contents of the 12 compounds among the samples, particularly for stem bark during the summer season, which showed higher contents of the 12 secondary metabolites that did that of the autumn season. This could be a vital indicator for chemotaxonomy. This systematic microscopic observation and phytochemical analysis provide scientific criteria to ensure the identity, quality, purity, and establishment of standard parameters for the safe use of medicinal plants. In addition, the *Fraxinus* plants and their isolates could be effective against obesity by decreasing fat absorption and accumulation through inhibition of pancreatic lipase. Further study is required to determine the exact underlying mechanism.

## Figures and Tables

**Figure 1 plants-09-00534-f001:**
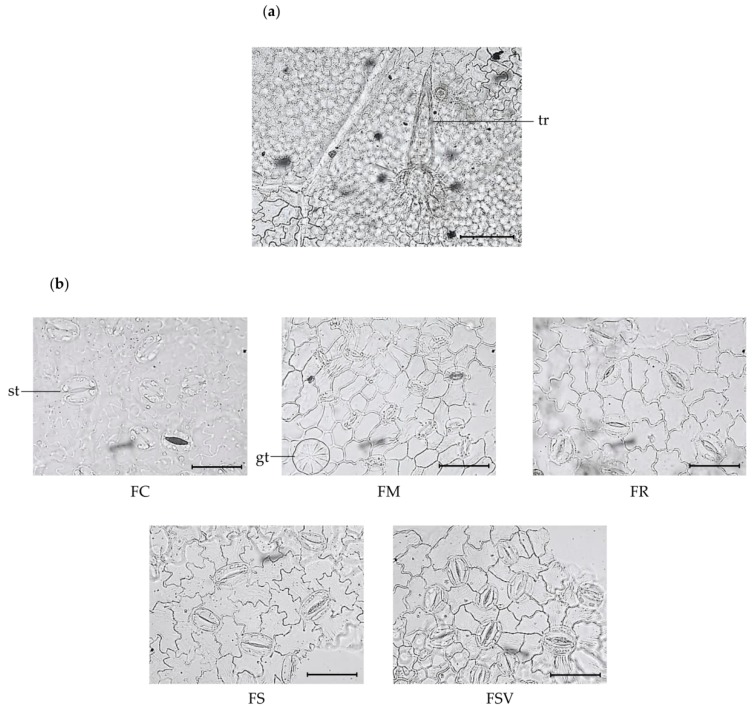
Photomicroscopic data of the leaf surface of five *Fraxinus* plants: (**a**) epidermal cells are arranged in a rosette around trichome base on the lower surface (*F. rhynchophylla*) (×400). The black bar means 100 μm; (**b**) photomicroscopic data of the lower surface of leaves from five plants (×1000). Black bars mean 50 μm; *gt*, glandular trichome; *st*, stomata; *tr*, trichome; FC, *Fraxinus chiisanensis*; FM, *F. mandshurica*; FR, *F. rhynchophylla*; FS, *F. sieboldiana*; FSV, *F. sieboldiana* var. *angustata*.

**Figure 2 plants-09-00534-f002:**
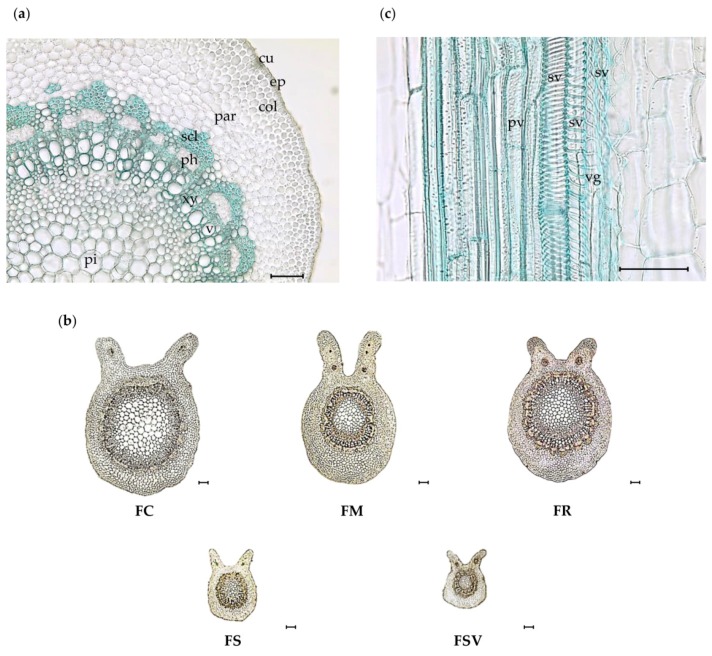
Photomicroscopic data of the petiole from five *Fraxinus* plants: (**a**) anatomical elements of petiole in transverse section (×200); (**b**) transverse section of the petiole of leaves from five *Fraxinus* plants (×40); (**c**) different vessels of petiole in vertical section (×400). Black bars mean 100 μm; *col*, collenchyma cell; *cu*, cuticle; *ep*, epidermis; *par*, parenchyma cell; *ph*, phloem; *pt*, pith; *pv*, pitted vessel; *s*, starch grain; *scl*, sclerenchyma cell; *sv*, spiral vessel; *vg*, annular vessel; FC, *Fraxinus chiisanensis*; FM, *F. mandshurica*; FR, *F. rhynchophylla*; FS, *F. sieboldiana*; FSV, *F. sieboldiana* var. *angustata*.

**Figure 3 plants-09-00534-f003:**
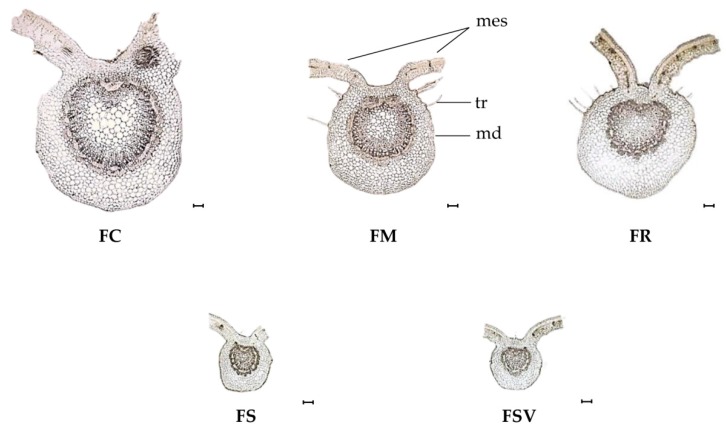
Transverse section of the midrib of leaves from four *Fraxinus* species and one variety (×40). Black bars mean 100 μm; *mes*, mesophyll; *md*, midrib; *tr*, trichome; FC, *Fraxinus chiisanensis*; FM, *F. mandshurica*; FR, *F. rhynchophylla*; FS, *F. sieboldiana*; FSV, *F. sieboldiana* var. *angustata*.

**Figure 4 plants-09-00534-f004:**
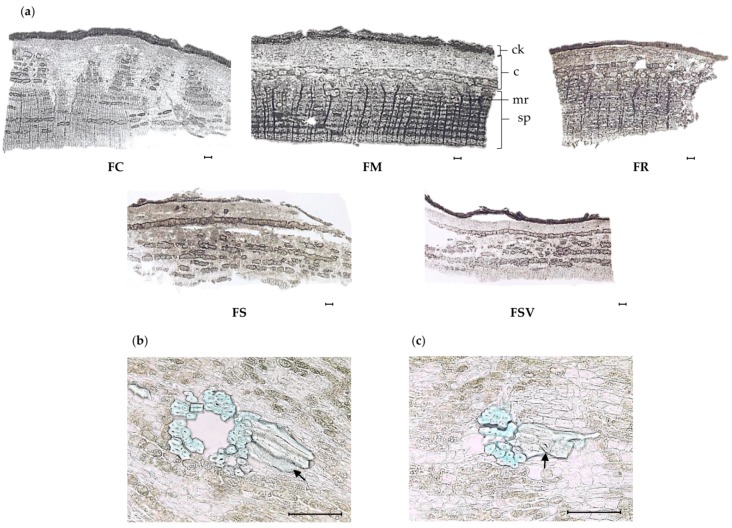
Photomicroscopic data of the stem bark of *Fraxinus* plants: (**a**) transverse section of the stem bark of four *Fraxinus* species and one variety (×40). Black bars mean 100 μm; (**b**) idioblastic; (**c**) brachysclereids sclereids of *F. mandshurica* (×400). Black bars mean 100 μm; *ck*, cork; *c*, cortex; *mr*, medullary ray; *sp*, secondary phloem; FC, *Fraxinus chiisanensis*; FM, *F. mandshurica*; FR, *F. rhynchophylla*; FS, *F. sieboldiana*; FSV, *F. sieboldiana* var. *angustata*.

**Figure 5 plants-09-00534-f005:**
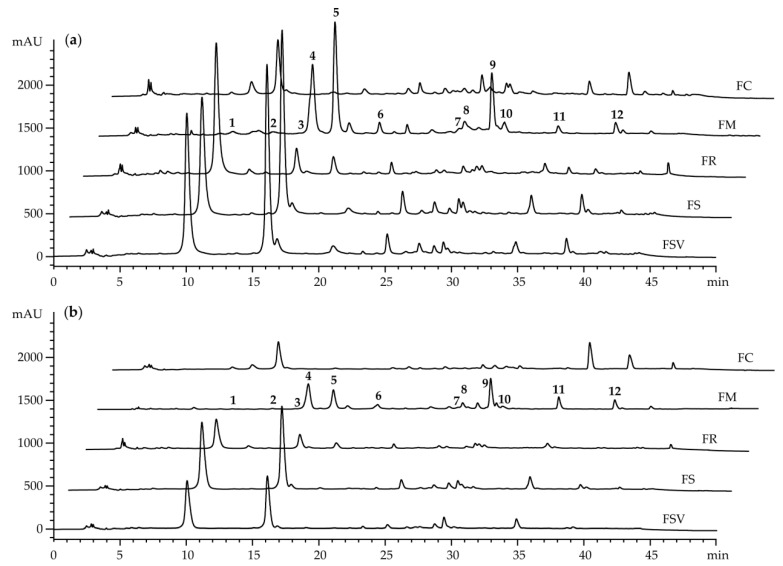
LC chromatograms of the extracts from the stem bark of five *Fraxinus* plants collected at summer (**a**) and autumn (**b**) seasons. FC, *Fraxinus chiisanensis*; FM, *F. mandshurica*; FR, *F. rhynchophylla*; FS, *F. sieboldiana*; FSV, *F. sieboldiana* var. *angustata*.

**Figure 6 plants-09-00534-f006:**
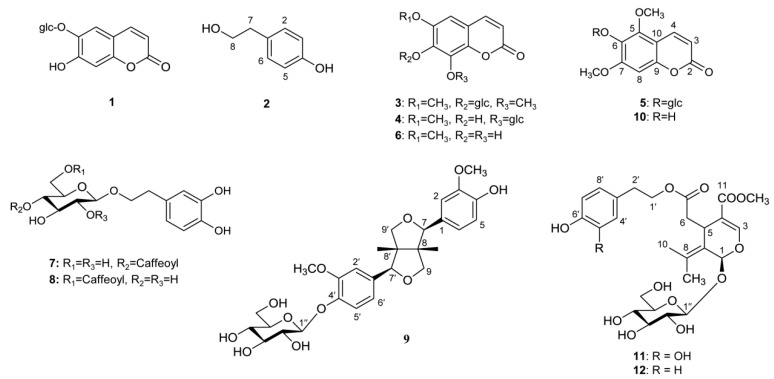
Chemical structures of peaks **1**–**12**.

**Figure 7 plants-09-00534-f007:**
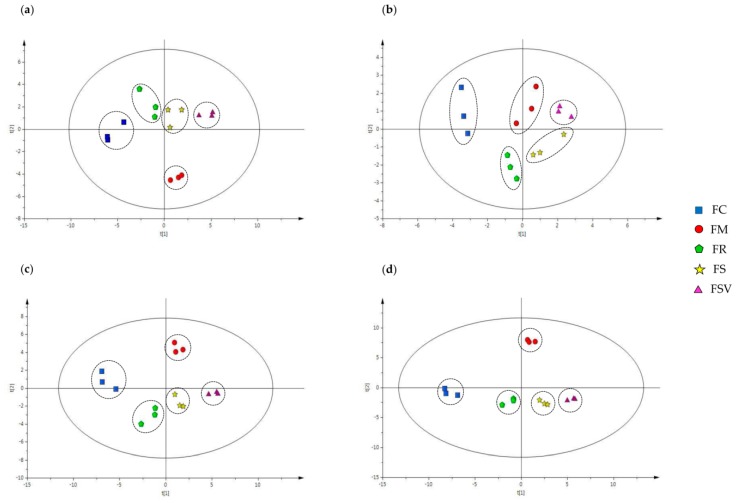
Orthogonal Projections to Latent Structures-Discriminant Analysis (OPLS-DA) multivariate statistical analysis of four *Fraxinus* species and variety: (**a**) anatomical data of leaves; (**b**) anatomical data of stem bark; (**c**) anatomical data of the leaves combined with anatomical data of stem bark; (**d**) anatomical data of leaf and bark combined with HPLC quantification data. FC, *Fraxinus chiisanensis*; FM, *F. mandshurica*; FR, *F. rhynchophylla*; FS, *F. sieboldiana*; FSV, *F. sieboldiana* var. *angustata*.

**Figure 8 plants-09-00534-f008:**
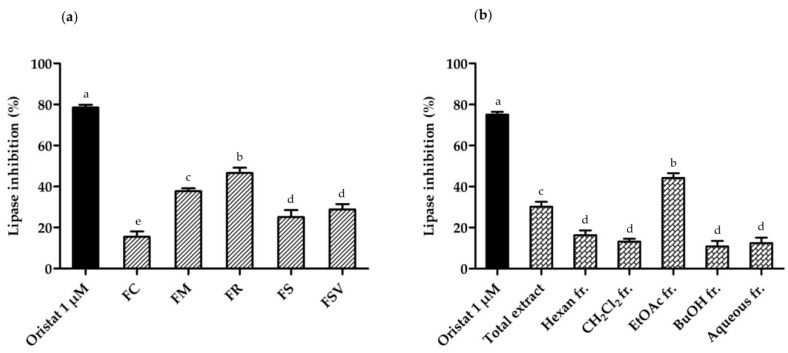
The anti-lipase activity of five *Fraxinus* plants and the fractions of FM: (**a**) percent lipase inhibition of five *Fraxinus* plants collected at summer season at 100 µg/mL; (**b**) percent lipase inhibition of total extract and five fractions from *F. mandshurica* bark at 100 µg/mL. Orlistat was used as a positive control. Different letters indicate significant differences based on one-way ANOVA followed by Tukey’s multiple comparison test (*p <* 0.05). FC, *Fraxinus chiisanensis*; FM, *F. mandshurica*; FR, *F. rhynchophylla*; FS, *F. sieboldiana*; FSV, *F. sieboldiana* var. *angustata*.

**Figure 9 plants-09-00534-f009:**
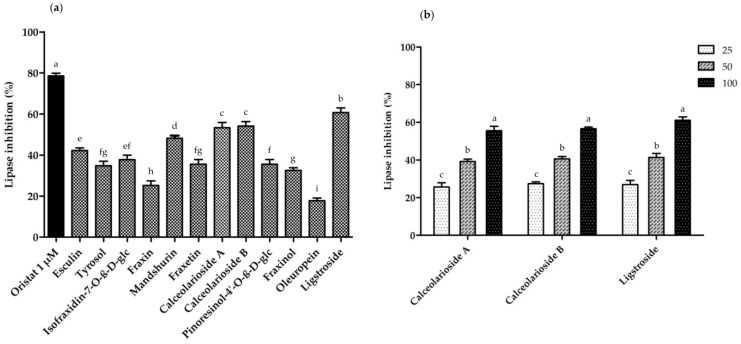
The anti-lipase activity of twelve compounds isolated from *F. mandshurica*: (**a**) percent lipase inhibition of the isolated compounds from *F. mandshurica* bark at the concentration of 100 μM. Orlistat was used as a positive control; (**b**) percent lipase inhibition of calceolarioside A and B, and ligstroside at three concentrations of 25, 50 and 100 μM. Different letters indicate significant differences based on one-way ANOVA followed by Tukey’s multiple comparison test (*p <* 0.05).

**Table 1 plants-09-00534-t001:** Anatomical characteristics of the lower leaf surface of *Fraxinus* plants.

Parameters	FC	FM	FR	FS	FSV
Number of glandular trichome (in 500 × 500 μm^2^)	1.7 ± 0.2^c^	2.4 ± 0.4^b^	2.5 ± 0.4^b^	3.6 ± 0.9^b^	7.0 ± 0.4^a^
Diameter of glandular trichome (μm)	41.1 ± 1.9^a^	42.8 ± 3.7^a^	36.3 ± 0.8^b^	42.3 ± 2.8^a^	32.3 ± 1.5^c^
Number of epidermal cells (in 300 × 300 μm^2^)	78.4 ± 11.3^c^	148.4 ± 12.5^a^	98.5 ± 14.2^c^	92.3 ± 4.8^c^	122.8 ± 8.2^b^
Number of stomata (in 300 × 300 μm^2^)	11.7 ± 0.4^e^	33.9 ± 4.0^a^	19.3 ± 1.8^c^	14.7 ± 1.3^d^	24.6 ± 3.0^b^
Stomatal length (μm)	37.5 ± 2.0^a^	22.8 ± 0.6^c^	33.4 ± 0.7^b^	34.7 ± 2.3^ab^	32.5 ± 1.6^b^
Stomatal width (μm)	26.0 ± 2.4^a^	17.1 ± 0.5^b^	23.9 ± 1.5^a^	24.2 ± 2.5^a^	25.0 ± 1.5^a^
Stomatal index	13.0 ± 1.1^b^	18.7 ± 1.5^a^	16.4 ± 1.8^a^	13.7 ± 1.3^b^	16.7 ± 0.9^a^

Values were expressed as the mean ± SD (*n* ≥ 9). The different upper letters in the same line indicate significant differences (*p* < 0.05) among the samples. FC, *Fraxinus chiisanensis*; FM, *F. mandshurica*; FR, *F. rhynchophylla*; FS, *F. sieboldiana*; FSV, *F. sieboldiana* var. *angustata*.

**Table 2 plants-09-00534-t002:** Anatomical characteristics of the petiole of *Fraxinus* plants.

Parameters	FC	FM	FR	FS	FSV
Size of epidermal cell of adaxial portion (length × width, μm)	17.4 ± 1.3^a^ × 17.8 ± 1^a^	11.2 ± 1^c^ × 11.8 ± 1^b^	19.4 ± 2.3^a^ × 15.6 ± 1.7^ab^	13.8 ± 1.3^b^ × 12.4 ± 2.2^b^	12.9 ± 1^b^ × 11.6 ± 0.9^b^
Size of epidermal cell of abaxial portion (length × width, μm)	13.3 ± 1.4^b^ × 13.6 ± 1.6^b^	10.2 ± 1.0^c^ × 11.5 ± 1^b^	21.1 ± 2.3^a^ × 18.7 ± 3.5^a^	17.5 ± 1.9^a^ × 14.9 ± 1.0^ab^	13.6 ± 0.9^b^ × 12.4 ± 0.8^b^
Ratio of cortex thickness to radius in abaxial portion	0.23 ± 0.03^b^	0.29 ± 0.01^a^	0.30 ± 0.04^a^	0.28 ± 0.02^a^	0.27 ± 0.01^a^
Diameter of parenchyma cell of cortex in adaxial portion (μm)	35.3 ± 2.5^a^	24.8 ± 1.9^b^	24.9 ± 2.0^b^	24.7 ± 1.5^b^	20.0 ± 0.4^c^
Diameter of parenchyma cell in abaxial portion (μm)	51.6 ± 4.6^a^	42.5 ± 1.8^b^	40.4 ± 2.7^b^	33.4 ± 2.0^c^	27.7 ± 1.2^d^
Number of sclerenchyma cell (in 300 × 300 μm^2^)	40.9 ± 5.3^d^	51.3 ± 3.2^c^	70.5 ± 4.7^b^	54.9 ± 3.0^c^	80.3 ± 3.4^a^
Diameter of parenchyma cell in pith (μm)	81.3 ± 2.6^a^	54.3 ± 3.0^b^	54.5 ± 10.9^b^	37.7 ± 2.7^c^	30.9 ± 2.3^d^

Values were expressed as the mean ± SD (*n* ≥ 9). The different upper letters in the same line indicate significant differences (*p* < 0.05) among the samples. FC, *Fraxinus chiisanensis*; FM, *F. mandshurica*; FR, *F. rhynchophylla*; FS, *F. sieboldiana*; FSV, *F. sieboldiana* var. *angustata*.

**Table 3 plants-09-00534-t003:** Anatomical characteristics of the midrib of *Fraxinus* plants.

Parameters	FC	FM	FR	FS	FSV
Size of epidermal cell of adaxial portion (length × width, μm)	21.8 ± 1.5^a^ × 19.1 ± 1.6^a^	17.0 ± 2.5^b^ × 14.9 ± 0.3^b^	21.8 ± 4.0^ab^ × 16.6 ± 2.1^ab^	15.0 ± 1.1^b^ × 13.5 ± 1.6^b^	13.1 ± 0.4^c^ × 11.9 ± 0.9^b^
Size of epidermal cell of abaxial portion (length × width, μm)	13.6 ± 1.1^ab^ × 14.3 ± 1.0^b^	11.3 ± 1.8^b^ × 13.6 ± 1.4^b^	16.3 ± 2.0^a^ × 18.8 ± 2.4^a^	13.1 ± 1.0^b^ × 15.1 ± 2.2^ab^	12.4 ± 0.6^b^ × 13.2 ± 0.5^b^
Number of sclerenchyma cell (in 300 × 300 μm^2^)	38.6 ± 3.0^c^	52.7 ± 9.2^b^	70.1 ± 6.5^b^	75.5 ± 17.1^b^	114.8 ± 18.7^a^
Diameter of parenchyma cell in adaxial portion (μm)	28.4 ± 1.9^a^	23.8 ± 1.4^ab^	29.9 ± 5.4^a^	20.7 ± 2.3^b^	18.1 ± 0.8^b^
Diameter of parenchyma cell in abaxial portion (μm)	53.4 ± 2.5^a^	43.8 ± 1.2^b^	45.2 ± 2.4^b^	35.0 ± 2.7^c^	29.2 ± 0.7^d^
Diameter of parenchyma cell in pith (μm)	75.7 ± 3.7^a^	61.4 ± 4.4^b^	49.2 ± 4.1^c^	28.4 ± 4.0^d^	22.2 ± 3.5^d^
Size of palisade cell (length × width, μm)	48.6 ± 4.6^a^ × 17.4 ± 4.7^a^	44.3 ± 4.5^ab^ × 12.5 ± 0.4^a^	44.4 ± 6.4^ab^ × 14.2 ± 0.7^a^	46.2 ± 3.0^a^ × 14.4 ± 1.5^a^	40.8 ± 1.8^b^ × 12.5 ± 0.5^a^
Palisade ratio	1.7 ± 0.1^b^	2.0 ± 0.2^a^	1.9 ± 0.2^ab^	2.1 ± 0.2^a^	2.2 ± 0.2^a^

Values were expressed as the mean ± SD (*n* ≥ 9). The different upper letters in the same line indicate significant differences (*p* < 0.05) among the samples. FC, *Fraxinus chiisanensis*; FM, *F. mandshurica*; FR, *F. rhynchophylla*; FS, *F. sieboldiana*; FSV, *F. sieboldiana* var. *angustata*.

**Table 4 plants-09-00534-t004:** Anatomical characteristics of the stem bark of *Fraxinus* plants.

Parameters	FC	FM	FR	FS	FSV
Size of cork cell (length × width, μm)	21.2 ± 3.4^ab^ × 53.6 ± 4.9^a^	20.1 ± 4.8^ab^ × 40.0 ± 3.6^b^	13.6 ± 1.3^b^ × 33.1 ± 2.3^c^	16.2 ± 0.9^b^ × 36.8 ± 1.8^bc^	22.1 ± 1.1^a^ × 36.2 ± 1.6^bc^
Diameter of parenchyma cell (in cortex length × width, μm)	28.1 ± 2.0^a^ × 55.3 ± 2.2^a^	24.8 ± 1.4^ab^ × 44.5 ± 1.9^b^	24.5 ± 2.7^ab^ × 51.6 ± 1.5^a^	23.0 ± 1.2^b^ × 42.9 ± 2.5^b^	22.8 ± 0.7^b^ × 40.6 ± 1.7^b^
Number of phloem fiber (in 500 × 500 μm^2^)	83.7 ± 4.6^c^	85.4 ± 3.2^c^	123.0 ± 16.5^ab^	138.4 ± 3.5^a^	122.7 ± 4.2^b^
Number of phloem ray (in 500 × 500 μm^2^)	3.4 ± 0.3^b^	4.5 ± 0.5^ab^	3.3 ± 0.2^b^	3.6 ± 0.5^b^	4.8 ± 0.2^a^
Size of phloem ray (length × width, μm)	40.5 ± 0.3^a^ × 21.1 ± 2.0^a^	34.8 ± 0.8^b^ × 15.0 ± 1.5^b^	34.5 ± 4.3^ab^ × 20.9 ± 1.2^a^	32.1 ± 4.5^b^ × 15.5 ± 2.1^b^	28.3 ± 0.4^b^ × 14.7 ± 0.4^b^

Values were expressed as the mean ± SD (*n* ≥ 9). The different upper letters in the same line indicate significant differences (*p* < 0.05) among the samples. FC, *Fraxinus chiisanensis*; FM, *F. mandshurica*; FR, *F. rhynchophylla*; FS, *F. sieboldiana*; FSV, *F. sieboldiana* var. *angustata*.

**Table 5 plants-09-00534-t005:** Contents of twelve compounds (**1**–**12**) in the stem bark of *Fraxinus* plants.

Compounds	Season	FC	FM	FR	FS	FSV
Esculin (1)	Summer	ND*	0.31 ± 0.04^c^	32.5 ± 2.4^b^	31.6 ± 2.5^b^	38.7 ± 3.1^a^
	Autumn	ND	0.24 ± 0.09^c^	24.8 ± 0.8^a^	28.8 ± 4.8^a^	20.8 ± 2.5^b^
Tyrosol (2)	Summer	ND	0.56 ± 0.11^a^	ND	ND	ND
	Autumn	ND	0.48 ± 0.09^a^	ND	ND	ND
Isofraxidin-7-*O*-*β*-D-glucopyranoside (3)	Summer	ND	2.39 ± 0.41^a^	0.23 ± 0.04^b^	ND	ND
	Autumn	ND	1.66 ± 0.14^a^	0.22 ± 0.01^b^	ND	ND
Fraxin (4)	Summer	ND	14.8 ± 2.2^b^	10.8 ± 1.4^c^	58.1 ± 3.9^a^	61.7 ± 3.9^a^
	Autumn	ND	8.32 ± 1.61^c^	ND	37.0 ± 2.4^a^	24.8 ± 3.0^b^
Mandshurin (5)	Summer	ND	20.6 ± 0.7^a^	ND	ND	ND
	Autumn	ND	10.2 ± 0.5^a^	ND	ND	ND
Fraxetin (6)	Summer	ND	0.27 ± 0.03^b^	0.08 ± 0.00^c^	1.39 ± 0.12^a^	1.53 ± 0.15^a^
	Autumn	ND	0.26 ± 0.01^a^	ND	ND	ND
Calceolarioside A (7)	Summer	10.9 ± 1.0^b^	4.26 ± 1.50^c^	2.23 ± 0.42^d^	13.5 ± 1.2^a^	10.7 ± 0.4^b^
	Autumn	7.44 ± 1.23^a^	1.96 ± 0.27^c^	ND	7.69 ± 1.42^a^	2.72 ± 0.53^b^
Calceolarioside B (8)	Summer	ND	30.4 ± 2.9^a^	10.3 ± 0.7^b^	10.6 ± 1.0^b^	9.11 ± 1.32^b^
	Autumn	ND	9.94 ± 1.45^a^	4.04 ± 0.27^b^	8.90 ± 1.74^a^	7.91 ± 1.13^a^
Pinoresinol-4′-*O*-*β*-D-glucopyranoside (9)	Summer	ND	12.1 ± 0.4^a^	0.62 ± 0.17^d^	1.74 ± 0.06^b^	0.82 ± 0.05^c^
	Autumn	ND	9.74 ± 0.56^a^	0.61 ± 0.06^c^	0.77 ± 0.04^b^	0.24 ± 0.00^d^
Fraxinol (10)	Summer	ND	0.53 ± 0.05^a^	ND	ND	ND
	Autumn	ND	0.47 ± 0.02^a^	ND	ND	ND
Oleuropein (11)	Summer	ND	16.9 ± 0.4^a^	10.2 ± 2.0^b^	5.34 ± 0.99^d^	6.77 ± 0.25^c^
	Autumn	ND	6.98 ± 0.32^a^	5.10 ± 0.97^b^	2.97 ± 0.59^c^	6.26 ± 0.95^ab^
Ligstroside (12)	Summer	27.9 ± 1.8^a^	15.5 ± 0.9^b^	2.11 ± 0.41^c^	2.74 ± 0.47^c^	1.97 ± 0.22^cd^
	Autumn	23.9 ± 0.8^a^	14.7 ± 0.7^b^	ND	2.51 ± 0.50^c^	1.80 ± 0.31^c^
Total coumarins	Summer	ND	38.9 ± 3.4^c^	43.6 ± 3.8^c^	91.1 ± 6.5^b^	101.9 ± 7.2^a^
	Autumn	ND	21.2 ± 2.3^d^	25.5 ± 0.8^c^	65.8 ± 7.2^a^	45.5 ± 5.5^b^
Total phenylethanoids	Summer	10.9 ± 1.0^d^	35.3 ± 4.5^a^	12.5 ± 1.1^d^	24.1 ± 2.2^b^	19.8 ± 1.7^c^
	Autumn	7.44 ± 1.23^c^	12.4 ± 1.7^b^	4.04 ± 0.27^d^	16.6 ± 3.2^a^	10.6 ± 1.7^bc^
Total secoiridoids	Summer	27.9 ± 1.8^b^	32.4 ± 1.3^a^	12.3 ± 2.4^c^	8.08 ± 1.46^d^	8.74 ± 0.47^d^
	Autumn	23.9 ± 0.8^a^	21.7 ± 1.0^b^	5.08 ± 0.97^d^	5.48 ± 1.09^d^	8.06 ± 1.26^c^
Total	Summer	38.8 ± 2.8^c^	118.6 ± 9.6^a^	69.1 ± 7.5^b^	125.0 ± 10.2^a^	131.3 ± 9.4^a^
	Autumn	31.3 ± 2.0^c^	65.0 ± 5.7^b^	34.8 ± 2.1^c^	88.6 ± 11.5^a^	64.5 ± 8.4^b^

Data were expressed as mean ± SD (mg/g dry weight) (*n* = 4). Different upper letters in the same line indicate significant differences (*p* < 0.05) among samples. FC, *Fraxinus chiisanensis*; FM, *F. mandshurica*; FR, *F. rhynchophylla*; FS, *F. sieboldiana*; FSV, *F. sieboldiana* var. *angustata*. *ND, not detected.

**Table 6 plants-09-00534-t006:** A list of *Fraxinus* plants collected from Korea.

Botanical Name	Collection Places (Latitude, Longitude)	Collection Year	Specimen No.	Abbreviations
*Fraxinus chiisanensis* Nakai	Sancheong (35°27′55.9″N, 127°56′06.7″E) (35°27′55.4″N, 127°56′07.9″E) Jiri Mt. (35°14′58.8″N, 127°42′31.2″E) (35°14′58.0″N, 127°42′29.8″E) Pocheon (37°45′07.1″N, 127°09′58.7″E)	2016–2018	PGSC-560-1~7	FC
*F. mandshurica* Rupr.	Sancheong (35°14′00.1″N, 127°47′53.9″E) Jinju (35°10′53.3″N, 128°05′42.8″E) (35°13′59.9″N, 127°47′54.1″E) Pocheon (37°44′57.2″N, 127°09′55.7″E)	2016–2018	PGSC-561-1~12	FM
*F. rhynchophylla* Hance	Jinju (35°10′53.1″N, 128°05′42.6″E) Jiri Mt. (35°15′02.1″N, 127°42′30.1″E) Youngwol (37°12′18.4″N, 128°26′05.3″E) Pocheon (37°45′10.2″N, 127°09′49.6″E)	2016–2018	PGSC-562-1~20	FR
*F. sieboldiana* Blume	Jiri Mt. (35°14′54.5″N, 127°42′18.3″E) Miryang (35°34′06.6″N, 128°59′34.4″E) (35°34′04.7″N, 128°59′34.9″E) Pocheon (37°45′11.9″N, 127°09′54.9″E)	2016–2019	PGSC-563-1~15	FS
*F. sieboldiana* var. *angustata* Blume	Jiri Mt. (35°14′53.7″N, 127°42′30.0″E) Miryang (35°34′05.2″N, 128°59′34.4″E) (35°34′05.7″N, 128°59′31.8″E) Sancheong (35°17′46.8″N, 127°58′29.6″E) Pocheon (37°45′11.4″N, 127°09′56.2″E)	2016–2019	PGSC-564-1~15	FSV

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
