# Peer review of "Comparative Studies of Fraxinus Species from Korea Using Microscopic Characterization, Phytochemical Analysis, and Anti-Lipase Enzyme Activity"

_plants, 2020, doi:10.3390/plants9040534_

Round 1
Reviewer 1 Report
This manuscript is well written and presents a mix of histology and chemical characterization in order to be able to identify a source of chemical compounds.
I've got few suggestions to improve this document.
Line 64: suppress "of" before "all peaks"
Line 71, the transition from the previous paragraphe to this one is not easy, should be ameliorated
In all figures and tables concerned I'll include in the legend the signification of FC, FM, FR, FS and FSV
Line 89, it's strange to cite [26,27] in this results section without other indication to the references, this should be stated in the discussion section.
Line 295 Add a "s" to "stem bark"
Line 300: "FSV displayed characteristic the..." consider rewritting of this part.
Author Response
Response to reviewer #1:
Thank you very much for your consideration. We have considered your comments (Original Reviewer’s Comments, ORC) very closely in revising this manuscript as follows.
ORC: Line 64: suppress "of" before "all peaks"
- The “of” was removed before “all peaks” (line 64).
ORC: Line 71, the transition from the previous paragraph to this one is not easy, should be ameliorated In all figures and tables concerned I'll include in the legend the signification of FC, FM, FR, FS and FSV
- In order to make the text that flows better, the whole paragraph of "Furthermore, pancreatic lipase … lipase [24,25].” was changed into the following paragraph of "Meanwhile, FR and its isolated compounds were reported to possess anti-obesity characteristics via inhibition of pancreatic lipase [21,22]. Pancreatic lipase is the major enzyme that converts ingested triglycerides to fatty acids, and as such decreases fat absorption through pancreatic lipase inhibition and can be useful to treat obesity [23,24]. Many countries have used medicinal plants as dietary supplements for the controlling of body weight, and thus natural inhibitors of pancreatic lipase could be potential candidates to control obesity [25]." (lines 71-76).
- The full description for the five abbreviations of FC, FM, FR, FS and FSV was added in all related Tables and Figures.
ORC: Line 89, it's strange to cite [26,27] in this results section without other indication to the references, this should be stated in the discussion section.
- The citation of references [26,27] was removed.
ORC: Line 295 Add a "s" to "stem bark"
- The “stem bark” was corrected into “stem barks” (line 309).
ORC: Line 300: "FSV displayed characteristic the..." consider rewriting of this part.
- The sentence of "FSV displayed characteristic the..." was changed into the following sentence of " FSV displayed the highest frequency of glandular trichomes on the lower leaf surface and the highest number of sclerenchyma cells in the petiole and midrib among the samples." (lines 314-316).
Thank you again.

Reviewer 2 Report
The paper “Comparative studies of Fraxinus species from Korea using microscopic characterization, phytochemical analysis, and anti-lipase enzyme activity” describes
systematic microscopic observations and secondary metabolite quantification of the Fraxinus species and one variety that have been utilized as a folk medicine in Korea. In addition, the goal of their study was the evaluation the anti-lipase activity.
The work is well designed, and the results are well discussed too.
The paper deserves some minor revisions.
I would suggest the author inserting the chemical structures of the molecules identified in peaks 1-12 in the text instead of reporting them in the Supplementary material. The discussion section describes very well the differences in the lipase inhibitory activity related to the chemical structures, and a look at them makes the reading clearer.
The author should explain in both the text and the material and method section the data related to the fractionation of F. mandshurica bark. It’s not clear if the authors obtained five different extracts from F. mandshurica bark with 5 different solvents (EtOAc, CH2Cl2, Hexan, H2O and BuOH) or if the authors obtained 5 fractions from a chromatographic separation. In each case, authors should describe the different extraction/fractions with all the details and the yields.
Author Response
Response to reviewer #2:
Thank you very much for your careful and thorough reading of this manuscript and for the thoughtful comments and constructive suggestions, which helps to improve the quality of this manuscript. We have considered your comments (Original Reviewer’s Comments, ORC) very closely in revising this manuscript as follows.
ORC: I would suggest the author inserting the chemical structures of the molecules identified in peaks 1-12 in the text instead of reporting them in the Supplementary material. The discussion section describes very well the differences in the lipase inhibitory activity related to the chemical structures, and a look at them makes the reading clearer.
- The chemical structures were included as Figure 6 in the main text not in the Supplementary material (in Page 9, line 224).
ORC: The author should explain in both the text and the material and method section the data related to the fractionation of F. mandshurica bark. It’s not clear if the authors obtained five different extracts from F. mandshurica bark with 5 different solvents (EtOAc, CH2Cl2, Hexane, H2O and BuOH) or if the authors obtained 5 fractions from a chromatographic separation. In each case, authors should describe the different extraction/fractions with all the details and the yields.
- In order to make the description about the fractionation of mandshurica bark clear, the precise procedure with the yield (%, w/w) was added in Materials and Methods Section as the following sentences: "The filtrate was dried under a stream of N2 gas and then used in pancreatic lipase assay. The air-dried bark of FM (960 g) was ground and extracted in sonication with 100% methanol. The extract was filtered and concentrated in vacuo. The resulting crude extract (187 g) was suspended in water and fractionated with n-hexane, dicholomethane, ethyl acetate and n-butanol, successively to yield n-hexane fr. (6.4%, w/w), CH2Cl2 fr. (3.2%), EtOAc fr. (32.1%), n-BuOH fr. (47.1%) and aqueous fr. (7.5%) fractions, respectively" (in Page 17, lines 409-414).
Thank you again.
